# Biomarkers of Low-Level Environmental Exposure to Benzene and Oxidative DNA Damage in Primary School Children in Sardinia, Italy

**DOI:** 10.3390/ijerph18094644

**Published:** 2021-04-27

**Authors:** Ilaria Pilia, Marcello Campagna, Gabriele Marcias, Daniele Fabbri, Federico Meloni, Giovanna Spatari, Danilo Cottica, Claudio Cocheo, Elena Grignani, Fabio De-Giorgio, Pierluigi Cocco, Ernesto d’Aloja

**Affiliations:** 1Department of Medical Sciences and Public Health, University of Cagliari, Monserrato, 092109 Cagliari, Italy; mam.campagna@gmail.com (M.C.); gabriele.marcias@libero.it (G.M.); daniele.fabbri@hotmail.it (D.F.); federicomeloni@hotmail.it (F.M.); pcocco@unica.it (P.C.); dsmsp@unica.it (E.d.); 2Department of Biomedical and Dental Sciences and Morpho-Functional Imaging, University of Messina, 98122 Messina, Italy; gspatari@unime.it; 3Environmental Research Centre, Istituti Clinici Scientifici Maugeri, 27100 Pavia, Italy; danilo.cottica@icsmaugeri.it (D.C.); claudio.cocheo@fsm.it (C.C.); elena.grignani@fsm.it (E.G.); 4Department of Health Care Surveillance and Bioethics, Section of Legal Medicine, Università Cattolica del S. Cuore, 00168 Rome, Italy; fabio.degiorgio@unicatt.it; 5Fondazione Policlinico Universitario A. Gemelli IRCCS, Università Cattolica del Sacro Cuore, 00168 Rome, Italy

**Keywords:** benzene, environmental exposure, children, biological monitoring, genotoxicity

## Abstract

Background: The main anthropic sources of exposure to airborne benzene include vehicular traffic, cigarette smoke, and industrial emissions. Methods: To detect early genotoxic effects of environmental exposure to benzene, we monitored environmental, personal, and indoor airborne benzene in children living in an urban area and an area near a petrochemical plant. We also used urinary benzene and S-phenylmercapturic acid (S-PMA) as biomarkers of benzene exposure and urinary 8-hydroxydeoxyguanosine (8-OHdG) as a biomarker of early genotoxic effects. Results: Although always below the European Union limit of 5 μg/m^3^, airborne benzene levels were more elevated in the indoor, outdoor, and personal samples from the industrial surroundings compared to the urban area (*p* = 0.026, *p* = 0.005, and *p* = 0.001, respectively). Children living in the surroundings of the petrochemical plant had urinary benzene values significantly higher than those from the urban area in both the morning and evening samples (*p* = 0.01 and *p* = 0.02, respectively). Results of multiple regression modelling showed that age was a significant predictor of 8-OHdG excretion, independent of the sampling hour. Moreover, at the low exposure level experienced by the children participating in this study, neither personal or indoor airborne benzene level, nor personal monitoring data, affected 8-OHdG excretion. Conclusions: Our results suggest the importance of biological monitoring of low-level environmental exposure and its relation to risk of genotoxic effects among children.

## 1. Introduction

Airborne benzene is a well-known ubiquitous volatile organic compound (VOC) that is still widespread in both environmental and occupational settings and is perhaps one of the most studied environmental pollutants [1]. Industrial emissions, cigarette smoke, and vehicular traffic are the main anthropic sources of environmental exposure to benzene [2]. Duarte-Davidson et al. (2001) estimated a daily benzene intake of 400 µg in regular smokers of 20 cigarettes per day. With regard to industrial sources, several studies have investigated low-level exposure to benzene in petrochemical workers and in populations living near petrochemical plants [1,2,3,4,5,6,7,8,9,10,11,12,13,14,15].

The International Agency for Research on Cancer (IARC) has classified benzene as a group 1 human carcinogen, with the compound being implicated in the development of myeloid leukemia and probably in the etiology of some lymphoma subtypes [16,17]. As benzene follows first-order kinetics, its genotoxic effects may occur by oxidative stress, and epigenetic mechanisms may come into play even at low exposure levels [9]. Consistently, environmental exposure has been related to a significant increase in risk among the general population [16,18], especially in hyper-susceptible groups, such as children [18,19]. The United States Environmental Protection Agency (U.S. EPA) estimated an excess risk of cancer in the general population of 1 × 10^−6^ as a consequence of a lifetime exposure to airborne benzene ranging 0.13–0.45 µg/m^3^ [20]. Moreover, World Health Organization (WHO) guidelines set a similar excess risk for lifetime exposure to airborne benzene of 0.17 µg/m^3^ [21].

National and international agencies have established threshold values for occupational and environmental exposure to benzene by taking into account toxicological and socioeconomic issues. For instance, the European Commission set an annual airborne threshold of 5 µg/m^3^ for the general population and a maximum concentration in automotive fuels of 1% in volume [19]; the European Scientific Committee on Occupational Exposure Limits (SCOEL) established a 3.25 mg/m^3^ Threshold Limit Value–Weighted Average (TLV-TWA) [22]. On the other hand, the U.S. National Institute for Occupational Safety and Health (NIOSH) has proposed a TLA-TWA in occupational settings of 0.32 mg/m^3^ [23].

However, regulatory limits might not be enough to nullify the health effects of benzene exposure, both in workers and the general population. Therefore, interest has grown in the evaluation of health risks related to low-level benzene exposure in populations living near industrial settlements. Several studies have measured airborne benzene by passive sampling [7,8,24]; other studies have biomonitored benzene exposure in potentially exposed subjects using urinary benzene and S-phenyl mercapturic acid (S PMA) as biomarkers of low-dose, chronic exposure [1,3,5]. Several other studies have correlated biomarkers of exposure to benzene with markers of DNA damage, such as urinary 8-hydroxydeoxyguanosine (8-OHdG) or DNA adducts [9,10,25]. In particular, DNA adducts were reported at significantly higher level in the nasal epithelium of 75 children living near a petrochemical plant compared to those of 73 children living in a rural area [11].

Our primary goal was to assess benzene exposure related to urban and industrial emissions in children from the largest urban area in the island of Sardinia, Italy, and from a nearby small town adjacent to a large petrochemical plant, where a cluster of childhood leukemia was previously reported [26]. We coupled environmental, personal, and indoor airborne benzene monitoring with biological monitoring, taking into account lifestyle confounders, such as parental cigarette smoking. Finally, to detect early genotoxic effects in relation to low-level airborne benzene, we measured urinary 8-OHdG excretion as a marker of oxidative stress to DNA.

## 2. Materials and Methods

### 2.1. Study Design and Studied Population

The present study was conducted from January 2015 to July 2016 in two locations of southern Sardinia (Italy): the metropolitan area of Cagliari, the main city and the regional capital, and Sarroch, a small town immediately bordering one of the largest oil refineries in Europe, approximately 20 km west from Cagliari. The area in between includes a vast wetland of environmental interest that is home to various avian species, such as nesting flamingos, and a flourishing shellfish farm. A container cargo ship terminal that is currently inactive, a large industrial area, and the major airport are in the immediate proximity. The industrial area included a large chemical plant, but the only functioning plant now is an electrolytic soda plant, which is fed by a large nearby salt farm; the rest of the plant was shut down in 2013. The climate conditions are typically Mediterranean and very similar in the two locations. The temperature, precipitation, and humidity range 5–15 °C, 11–17 mm, and 78–84% in wintertime, respectively, with corresponding figures of 17–28 °C, 0.3–1.5 mm, and 67–72% in summertime. Prevalent wind direction is from northwest at the average speed of 14 and 15 km/h in wintertime and summertime, respectively. Children between 3 and 13 years old attending primary school in three different locations of either area were eligible for the study. Table 1 shows selected features of the two areas.

The research team met with parents in each school to explain the objectives and the protocol of the study as well as promote participation on a voluntary basis, with no exclusion criteria apart from being healthy and not taking any medication. Overall, the families of 88 children, 40 (45.5%) from schools in the urban area and 48 (54.5%) from schools in the area near the industrial site, accepted participation. Residence of children was homogeneously distributed over each study area. Following signature on the informed consent form, parents of the participating children filled a self-administered questionnaire gathering information on sociodemographic conditions (number of cohabitants, outdoor or indoor location of their child’s physical activity, preferred type of transportation (public or private), area of residence (peripheral, central, or rural), type of dwelling, internal house surface area, floor level, number of windows, type of heating, and distance from the nearest gas station), smoking, and dietary habits. The questionnaire was accompanied by detailed instructions on how to sample urine and how to manage the passive sampling devices.

### 2.2. Personal and Environmental Sampling

We carried out personal, indoor, and outdoor airborne benzene samplings with Radiello^®^ passive, radial diffusive samplers (Centro Ricerche Ambientali—Istituti Clinici Scientifici Maugeri, Padova, Italy). This device contains an adsorbing cartridge (Carbograph 4, Centro Ricerche Ambientali—Istituti Clinici Scientifici Maugeri, Padova, Italy) covered with a cylindrical diffusive body and is mounted on an easy-to-wear polycarbonate support. We asked children to wear the diffusive sampler near the breathing zone for one week (from Friday to Friday) during all their activities. While sleeping or during sport activities, parents had to deposit the diffusive sampler as near as possible to the child. During the same week, a second diffusive sampler was placed in the living room or in the room where the child spends most of their time for indoor air monitoring.

Outdoor monitoring campaigns were conducted in both areas, the first in winter (January–March) 2015, the second in summertime of the same year, and the third in May–July 2016, during the personal and indoor monitoring campaigns. Twenty radial diffusive samplers (10 in Cagliari and 10 in Sarroch) were placed in selected points within the study areas following the indications of the Life RESOLUTION (Development of a high spatial resolution atmospheric monitoring model to verify the actual emissions reduction of ozone precursors foreseen by Auto-Oil program, Project Life 99ENV/IT/081) project, coordinated by the Environmental Research Center of the Istituti Clinici Scientifici Maugeri (Padova, Italy) [30]. The sampling protocol followed the European Union guidelines for urban pollution assessment [31]. During outdoor monitoring, the diffusive sampler was protected against bad weather with a polypropylene box.

Each Friday morning, the research team collected the radial diffusive samplers; the adsorbing cartridges were capped in their specific glass tubes at the controlled temperature of 4 °C before shipment to the Environmental Research Centre of Istituti Clinici Scientifici Maugeri (Padova, Italy) for analysis.

As benzene and toluene are among the main constituents of vehicular exhausts, airborne toluene was also measured in all the airborne passive samplings to calculate the toluene: benzene (T/B) ratio as an indicator of the contribution of traffic emissions [32].

### 2.3. Urine Sampling

At the end of the week of personal and indoor air monitoring, each child provided two urine samples in polyethylene containers. Parents were instructed to collect two urine samples from their child on the last two days of the weekly monitoring, one on Thursday evening and the second on Friday morning before breakfast, and to transfer a 5 mL urine aliquot immediately after collection in a hermetically sealed glass vial using a syringe for benzene analysis. The remainder was used for detection of creatinine, S-PMA, cotinine, and 8-OHdG. Evening samples had to be stored at 4 °C during nighttime; on Friday morning, the research team collected all the samples at school and transported them to the Cagliari University laboratories for immediate storage at −20 °C until shipment in dry ice to the Environmental Research Centre of Istituti Clinici Scientifici Maugeri (Pavia, Italy) for analysis.

### 2.4. Chemicals and Supplies

The Radiello^®^ passive, radial diffusive samplers were purchased from Istituti Clinici Scientifici Maugeri, Padova, Italy. Benzene (≥99.9%), toluene (≥99.9%), the corresponding deuterated standards benzene-d_6_ and toluene-d_8_ used as internal standards, cotinine (purity 98%), cotinine-d_3_ (purity 99%), s-phenylmercapturic acid (S-PMA, purity 98%), and 8-hydroxy d-guanosine (8-OHdG, purity 98%) were purchased by Sigma-Aldrich (Milan, Italy). All standards were used without further purification. Hight Performance-Liquid Chromatography (HPLC)-grade water and methanol were from Carlo Erba (Milan, Italy). Stock solutions containing about 1 g/l of benzene, toluene, and S-PMA were prepared in methanol; cotinine was dissolved in water.

### 2.5. Analytical Methods

#### 2.5.1. Determination of Airborne Benzene and Toluene

Benzene and toluene analysis from passive diffusive samplers was conducted by gas chromatography coupled by mass spectrometry (GC-MS, 6890-5973, Agilent, Santa Clara, CA, USA) with a 70 eV inert electron impact source operating in single ion monitoring (SIM) mode and a liquid auto-injector (Agilent 7638, Cernusco sul Naviglio, Italy), following automatic thermal desorption (TurboMatrix Thermal Desorber, Perkin Elmer, Shelton, UK). We used a DB1 capillary column (60 m, 0.25 mm i.d., 1.0 μm film thickness; J&W Scientific, CPS Analitica, Milan, Italy) for analyte separation. The analysis was conducted with helium as the carrier gas at a constant flow rate of 1 mL/min and injector temperature of 250 °C. The GC oven temperature was as follows: from 40 °C (10 min) to 90 °C at 10 °C/min, hold 3 min; then to 120 °C at 20 °C/min, hold 2 min; and finally to 160 °C at 30 °C/min, hold 2 min. Benzene and toluene retention time was 14.95 and 19.70 min, respectively. Signals were acquired in SIM mode registering the positive ion to charge ratio *m*/*z* as follows: 78 benzene and 91 for toluene. The limit of detection (LOD) was 0.05 µg/m^3^ for both analytes.

#### 2.5.2. Determination of Biological Markers in Urines

Urinary benzene was determined by headspace solid-phase microextraction (SPME, CAR/PDMS 75 µM, Supelco, Bellefonte, PA, USA) followed by GC–MS (MSD 5973, Agilent, Santa Clara, CA, USA) using a previously published method [33,34]. The limit of detection (LOD) was 0.02 µg/L, and coefficient of variation (%CV) was 9%.

The determination of urinary S-PMA was performed with an Acquity ultra performance liquid chromatography (UPLC) system coupled with a triple quadrupole mass spectrometer (UPLC–MS–MS, Waters, Milford, MA, USA) using a recently published modified method [35]. For chromatographic separation, we used a UPLC HSS C18 column (2.1 × 150 mm, 1.7 μm). The method is linear in a concentration range of 1.0–25 µg/L. The LOD was 0.03 µg/L.

For 8-OHdG assays, we also used UPLC–MS–MS (Waters, Milford, MA, USA) and chromatographic separation on a UPLC BEH C18 column (2.1 × 100 mm^2^, 1.7 μm) following the bioanalytical method guideline of the European Medicines Agency [36]. The LOD for 8-OHdG was 1.5 nM.

Urinary cotinine was analyzed using a triple quadrupole GC–MS (7000A GC-QQQ Series MSD, Agilent, Santa Clara, CA, USA) [37]. The LOD was 0.20 ng/mL with a <5% CV.

To correct S-PMA and cotinine values for urinary creatinine excretion, we used the Jaffe’s colorimetric method to assess urinary creatinine by volume (g/L) [38]. In agreement with the World Health Organization guidelines [39], we excluded five samples with creatinine readings <0.3 or >3 g/L, all from the urban group.

Measurements below LOD (63% of urinary benzene and 11% S-PMA in the morning samples and 59% of urinary benzene and 6% S-PMA in the evening samples) were replaced with half the LOD [40]. There were six subjects with missing values in both morning and evening samples in all urinary biomarkers, and they were replaced with the respective median value among all subjects of the same gender and area. Missing values in one sample only of the coupled morning/evening samples (*n* = 21) were replaced with the predicted value obtained from the regression formula with the respective morning or evening sample as the independent variable among the available readings [41].

#### 2.5.3. Statistical Methods

We assessed the normal distribution of the variables with the Kolmogorov–Smirnov test. For parametric, normally distributed variables, we used mean and standard deviation as the descriptive statistics and the Student’s *t* test for independent series and analysis of variance (ANOVA) to compare differences between study groups. For the air monitoring data, we used median and interquartile (IQ) range as the descriptive statistics and the Mann–Whitney and the Wilcoxon statistics to test the differences between study groups and by time of the day, respectively. We used the Spearman’s correlation coefficient to assess the correlation between airborne concentrations, biomarkers of exposure, cotinine, and markers of oxidative stress to DNA. To explore the correlation between passive smoking (mostly from parental smoking) and children’s urinary cotinine, we created a binary variable expressing whether or not there were smoking cohabitants.

To overcome the analytical difficulties created by the elevated proportion of urinary benzene measurements below the LOD, we created two binary variables for urinary benzene, one using the LOD as the cut point and another using the top (fourth) quartile. For the sake of comparison with urinary benzene, we also created a binary variable for S-PMA using the top (fourth) quartile as the cut point.

We conducted multiple regression analysis to identify which variables were most predictive of the S-PMA and the 8-OHdG readings in both sampling times.

We also designed logistic regression models to predict urinary benzene values above the LOD and above the top (fourth) quartile, respectively. Regression models included age, gender, number of cohabitants, number of smoking cohabitants, residence in industrial vs. urban area, location in the area (central vs. peripheral), presence of a gas station near the residence, internal surface size of the residence, floor level, indoor and personal airborne benzene values, and urinary cotinine. Variables that significantly reduced the residual variance were retained in the final models.

We rejected the null hypothesis when it was associated with a *p* value less than 5%. The analysis was conducted with SPSS (version 20.0 package for Windows, SPSS Inc., Chicago, IL, USA) and SAS^®^ Studio (University Edition).

## 3. Results

As shown in Table 2, there were no substantial differences between the participating children in the two study areas in terms of age, gender, family size, and number of smoking cohabitants.

Airborne concentration of benzene and toluene and their ratio over all passive samplings are shown in Table 3 by study area. Except for the summer 2015 environmental sampling campaign, samples from the industrial area had significantly higher levels compared to the urban area. Airborne benzene concentrations during the 2015 sampling campaign were significantly more elevated in wintertime than in the summertime samplings in both locations (Wilcoxon test: *p* = 0.008). In 2015, airborne toluene level was significantly higher near the petrochemical plant in wintertime samplings (*p* = 0.008), while higher T/B ratios were observed in the urban area in the summertime sampling campaigns (*p* = 0.005).

Indoor and personal airborne benzene levels, but not toluene levels, were moderately though significantly elevated among children residing in the industrial area (Table 4). Measurement levels decreased significantly with increasing floor level but not by whether the residence was central, peripheral, or rural in the map of both the urban and industrial areas. Personal and indoor airborne measurements and urinary biomarkers also did not vary with the size of the residence, the number of windows, or the number of cohabitants (not shown in the tables).

There was a positive correlation between personal and indoor benzene measurements (Spearman’s correlation coefficient = 0.590, *p* = 4.37 × 10^−9^), and between personal benzene measurements and S-PMA urinary concentration (evening: *r* = 0.379, *p* = 0.0004; morning: *r* = 0.293, *p* = 0.007) but not urinary benzene (evening: *r* = 0.059, *p* = 0.594; morning: *r* = 0.094, *p* = 0.400). The correlation was weaker between indoor measurements and evening S-PMA (*r* = 0.244; *p* = 0.05).

Table 5 shows the median value and IQ range of biomarkers of exposure to benzene (urinary benzene and S-PMA), cigarette smoke (urinary cotinine), and oxidative damage to DNA (8-OHdG). Children from the industrial area had urinary benzene values significantly higher than the urban group (Mann–Whitney test: evening samples *p* = 0.01; morning samples *p* = 0.02). S-PMA was significantly more elevated in the evening samples in both the industrial and urban study areas (*p* < 0.0001), but it did not vary substantially by study area. 8-OHdG morning values were also higher among children from the industrial area (*p* = 0.02). Neither urinary benzene nor 8-OHdG levels varied by the time of the day.

In the overall study population and within both study areas, cotinine excretion increased with the presence of smoking cohabitants in both the evening samples (ANOVA, *F* = 7.36, *p* = 0.001) and the morning samples (ANOVA, *F* = 4.53, *p* = 0.01) and with the number of cigarettes smoked daily by the family members in the evening samples (Spearman’s correlation coefficient = 0.455 *p* = 0.008) but not in the morning samples (Spearman’s correlation coefficient = 0.137 *p* = 0.447). Cotinine values did not vary by time of the day (Wilcoxon test: *p* = 0.091,) nor did evening values vary by study group (Mann–Whitney test: *p* = 0.376), while morning values were significantly elevated among children from the urban area (Mann–Whitney test: *p* = 3.6 × 10^−12^).

Table 6 shows the results of the logistic regression analyses predicting urinary benzene level and urinary S-PMA level above the upper quartile in the samples collected in the evening hours. Covariates retained in the final regression models were those contributing to increasing the fitness of the models, as represented by the *R*^2^ value. In model 1, these were age and gender of children, area of residence (industrial vs. urban), central or peripheral location of the home address within the area, and floor level. The model 1 results show that residence in the industrial area was the strongest predictor of both elevated urinary benzene and urinary S-PMA excretion in the evening samples. Interestingly, residence in downtown areas was associated with both urinary benzene and S-PMA levels above the upper quartile, and residence at the floor level was associated with elevated benzene biomarker levels. Covariates in model II included age, gender, airborne benzene from personal samples, and urinary benzene above the fourth quartile. Airborne benzene from personal samplings was the only independent variable significantly affecting S-PMA excretion in both morning and evening samples, while none of the covariates in this model was predictive of elevated urinary benzene excretion.

Results of the regression model predicting evening urinary levels below or above the LOD were consistent (not shown in the tables); multiple regression models, including the same covariates to predict benzene and SPMA urinary excretion, both yielded results consistent with the logistic regression analysis (not shown in the tables).

Urinary 8-OHdG excretion was significantly related to indoor airborne benzene level (morning: *r* = 0.276, *p* = 0.01; evening: *r* = 0.228; *p* = 0.04) but not to personal monitoring data (morning: *r* = 0.140, *p* = 0.21; evening: *r* = 0.190; *p* = 0.09) (not shown in the tables). On the other hand, urinary S-PMA, but not benzene and cotinine levels, showed a correlation with 8-OHdG (evening: Spearman’s correlation coefficient = 0.222, *p* = 0.043; morning: Spearman’s correlation coefficient = 0.246, *p* = 0.025), (not shown in the tables). Age of children was also inversely related to urinary 8OHdG in both morning and evening samples. However, these associations were not confirmed by multiple regression analysis, adjusted by age, gender, and residence area, where a younger child’s age was the only significant predictor. Results were similar with urinary benzene as a binary variable with cut point at the top quartile (Table 7), cut point above LOD, or as a continuous variable (not shown in the tables).

## 4. Discussion

In the present study, we combined environmental monitoring and biological monitoring data to investigate residential exposure to benzene and the related genotoxic effects using a biomarker of oxidative DNA damage among children, a segment of the population especially susceptible to the detrimental effects of environmental pollutants [42]. In most samples, airborne benzene was below the limit of 5 μg/m^3^ established by the European Commission (EC) for the general population as a weighted average over a one-year monitoring period [19]. Levels exceeding the EC limit were detected in the following: (i) one 2015 wintertime outdoor sampling near the petrochemical plant (5.29 μg/m^3^); (ii) both indoor and personal monitoring of a child living near the petrochemical plant (5.93 and 5.47 μg/m^3^, respectively); (iii) one indoor sampling in the urban area (5.49 μg/m^3^). However, it is unclear whether the short sampling time (one week) in our study, although repeated with consistent results in summertime and wintertime, might have led to overestimating airborne benzene levels with higher levels in children living and attending schools near the petrochemical plant. The finding of increased airborne benzene levels in wintertime is consistent with previous reports [12].

The median benzene exposure level we detected with personal diffusive samplers was lower than that reported by Fustinoni et al. in the same areas [1]. However, those measurements were taken during the busy hours of a workday only, which could have led to further overestimating benzene exposure. Moreover, in contrast to our findings, those results did not show a difference between airborne benzene values near the petrochemical plant and those in the urban area of Cagliari [1], probably due to different location of residence of the participating individuals and lower number of participants (11 residents vs. 54 in our study).

Indoor benzene levels ranging 0.85–0.90 μg/m^3^, slightly lower than those we measured in our study, were detected in a school in Rome, with median values of 2.22–2.47 μg/m^3^ observed in two dwellings [43]. Ferrero et al. (2017) reported indoor benzene levels to be significantly higher than outdoor levels in an area with diverse sociodemographic and environmental traits [14], with higher benzene level detected in relation to indoor ambient emission sources, such as fireplaces, gas stoves, cleaning and painting products, and environmental tobacco smoke [7,44,45]. In our study, outdoor benzene levels exceeded indoor levels, and T/B ratios were more elevated in outdoor measurements, which would be in agreement with a study of three schools near major roadways, similar to the one we conducted, suggesting an influence of traffic emissions on the indoor microenvironment [46]. Previous studies in the area did not assess toluene and T/B ratios [1,11,12]. The T/B ratios in our study are comparable to those reported in Bari, a major urban area in southern Italy [47], further suggesting a role of proximity to traffic emissions as a major determinant of indoor benzene levels [48]. Other reports have suggested T/B ratios higher than 10 in proximity of industrial sources [49]. We observed a higher T/B ratio in the summertime outdoor samples from the urban area compared to the area surrounding the petrochemical plant, but not in the wintertime samples, and in the indoor samples of the urban area, which would fit with the hypothesis of urban traffic as a larger contributor to indoor exposure to benzene. In our study, we used urinary benzene and S-PMA as biomarkers of benzene exposure in children, which are considered to be more suitable to assess low-level environmental exposure to benzene than t,t-muconic acid (t,t-MA) [5,50,51,52]. The median urinary concentrations of benzene, S-PMA, and cotinine were in line with the reference values for nonsmokers in the Italian population [53].

The median urinary benzene was lower than that observed in the 2011–2012 National Health and Nutrition Examination Survey (NHANES) in 417 children aged 6–11 years [54]. Urinary benzene and S-PMA values in our study were comparable to those previously measured in adults and children living in rural and urban areas [5,9,55], and they were one to two orders of magnitude lower than those measured at the end of the work-shift in 86 nonsmoker Italian traffic policemen occupationally exposed to mean airborne benzene levels of 17.3 μg/m^3^ [56], which would further support the role of automobile traffic in low-level benzene exposure.

A role of petrochemical plant emissions in contributing to airborne benzene levels would be expected. In a previous study on the adult nonsmoking and nonoccupationally exposed population from the same two areas, the urinary benzene levels were in line with our results; however, in contrast to the present report, there was no difference by area (urban or near the petrochemical plant) [15]. Besides, in a third study on the same setting, urban residents had urinary benzene levels, but not S-PMA levels, higher than subjects residing near the petrochemical plant [1]. Consistent with reports of urinary benzene levels above the LOD almost exclusively among smokers [5], 62.6% of urinary benzene values in the morning samples from our infant study population were below the LOD; besides the different location of residence and the larger size of our study, the different age range and the larger proportion of undetectable levels might account for the discrepancy with previous results. Sample handling might also affect U-Benz monitoring results [57]. However, appropriate training of parents on sample collection, like the one applied in our study, would allow acceptable analytical performance [58].

A significant correlation between urinary excretion of benzene, toluene, ethylbenzene, and xylenes and urinary cotinine was observed in studies of adult smokers and passive smokers [59,60] but not in nonsmokers [1,61,62]. In our study, we did not observe a correlation with passive smoking in children. Direct exposure of children to secondhand smoking might have been a reason of concern in thoughtful parents, thus preventing urinary cotinine from matching the questionnaire information.

Biomarkers of exposure to benzene were not inter-related in our study, while in multiple regression models, personal benzene exposure was a strong predictor of S-PMA excretion, thus confirming it as predicting sensible biomarker of exposure to benzene [9,63].

Several biomarker studies have shown that exposure to benzene can result in oxidative DNA damage [64,65,66,67]. In particular, urinary S-PMA was found to affect 8-OHdG level [9]. Moreover, a previous cross-sectional study of school children from the same area near the petrochemical plant we studied in this paper reported a urinary excretion of malondialdehyde–deoxyguanosine and bulky DNA adducts in the nasal epithelium that was higher than in children from a rural area [11]. In that study, outdoor benzene concentrations were similar to our wintertime 2015 measurements. In our study, urinary S-PMA was correlated with 8-OHdG excretion, and there was a significant increase of 8-OHdG excretion with increasing urinary S-PMA level in the multiple regression analysis, adjusting by age, gender, and study area. Moreover, while residing near the petrochemical plant was a significant predictor of oxidative stress in the univariate analysis, which is consistent with previous reports [9], the same association did not show up in multiple regression analysis. We only observed a significant inverse association between children’s age and 8-OHdG excretion, which we tentatively interpret as being due to greater susceptibility to environmental pollutants among the youngest.

The differences between the results from the morning and evening samples might be related to the reduction of emission sources, such as urban traffic, business activities, and exposure to cigarette smoke at nighttime, as reflected by morning sampling [9].

With respect to the existing literature, our report explored the full pathway from multiple sources of exposure to benzene, monitored in wintertime and summertime through environmental outdoor and indoor measurements, personal measurements, and questionnaire information, to biomarkers of dose and effect. Previous publications did infer exposure by comparing results in proximity of the petrochemical plant with urban areas [1,9,11] or remote villages [11]. In our study, measurements of toluene and the T/B ratio allowed us to compare the contribution of industrial emissions and that from vehicular traffic. Other studies, conducted in other areas of mainland Italy or elsewhere in the world, limited the exposure assessment to biomonitoring data [9,13] or to outdoor and indoor environmental measurements [43]. A Turkish study of school children in a major urban area in proximity of several industrial settlements only focused on urban emissions [7]. On the contrary, a Brazilian study only considered exposure from a near petrochemical plant with very limited biomonitoring data [8]. Limitations in previous studies included considering only indoor benzene measurements in toddlers [14] and limiting the monitoring period to one season, mainly springtime [10] or summertime [9].

Our study was also affected by several limitations. First, we only used 8-OHdG; 8-oxo-7,8 dihydroguanosine (8-OHG) is reportedly the most sensitive biomarkers [68] of oxidative stress, and it would have been perhaps more appropriate for low-level environmental exposure to benzene. However, both 8-oxodGuo and 8-oxoguo are well correlated with oxidative stress [68,69,70], and in both evening and morning samples, 8-oxodGuo and 8-oxoGuo correlated with each other [9]. However, urinary 8-OHdG is one of the predominant forms of free-radical-induced oxidative lesions in nuclear and mitochondrial DNA [70]. There is evidence that DNA oxidation is associated particularly with cancer outcomes, while RNA oxidation, which is more typically indicated by 8-OHG level, seems more associated with neurodegenerative diseases and diabetes [71]. As we were interested in low-level benzene exposure as a possible determinant of childhood leukemia in the study area [26], we selected 8-OHdG as the biomarker of oxidative stress in our study. Our choice was consistent with other similar studies conducted in Thailand [10,72].

The large fraction of undetectable urinary benzene excretion levels was a reassuring result for the health of the participating children, but it was a further interpretative limitation for our findings. We followed standard procedures to replace urinary benzene values below LOD for the purposes of analysis [40]. We also used urinary S-PMA as a biomarker of dose, and we came up with the same results after transforming the benzene and S-PMA values into binary variables to avert the analytical problem. The study size was also a limitation in this study, although it was still larger than previous studies conducted in the same area. We suspect the complexity of the study protocol, the need for parents to follow precise instructions, and the requirement for children to wear the passive Radiello^®^ sampler all day long, including at school, might have discouraged participation. Besides, we did not have information on children’s direct exposure to parental smoking but only on the number of smoking cohabitants, which might not be relevant if parents were thoughtful enough to avoid smoking in the presence of their children.

## 5. Conclusions

In this study, airborne benzene was below the European Commission standards for the general population, and benzene biomarkers confirmed very low exposure levels. Still, in agreement with previous studies [73,74], we detected an association with oxidative DNA damage, although there was no association with biomarkers of dose. Overall, our findings contribute to the knowledge on the potential health effects of low-level exposure to benzene, and they might be useful for the development of governmental strategies of prevention.

## Figures and Tables

**Table 1 ijerph-18-04644-t001:** Resident population, motorization rate, and annual mean airborne level of benzene and PM2.5 in the two study areas.

Area Attributes	Cagliari (Urban)	Sarroch (Industrial)
**Resident population** * (No.)	154,083	5267
**Population density** * (No. per km^2^)	1821.9	77.7
**Motorization rate** ** (No. per 1000 inhabitants)		
Auto vehicles	671.1	not available
Motorcycles	96.3	not available
**Annual mean of airborne pollutants** ***		
Benzene (µg/m^3^)	1.9	1.6
PM2.5 (µg/m^3^)	16	13

References: * [27] ** [28] *** [29].

**Table 2 ijerph-18-04644-t002:** Resident population, motorization rate, and annual mean airborne level of benzene and PM2.5 in the two study areas.

Variable	Total	Urban	Industrial	*p*-Value
**Gender**	83	35	48	-
(numbers of male, female)	(37, 46)	(13, 22)	(24, 24)	
**Age**				0.20
Mean, SD	8.0, 2.3	7.6, 1.9	8.3, 2.5	
(min–max)	(3–13)	(3–11)	(3–13)	
**Family size**				0.80
Mean, SD	4.0, 1.0	4.0, 1.0	4.0, 1.0	
**Number of smokers in the household**				0.10 **
None	49	15	34	
1	17	10	7	
≥2	13	6	7	
Missing	4	4	0	
**Number of cigarettes/day ***				0.2
Mean, SD	13.2, 10.0	12.9, 10.8	13.5, 9.3	

* Among smokers only. ** chi-square test.

**Table 3 ijerph-18-04644-t003:** Airborne benzene and toluene concentrations (expressed as µg/m^3^) in outdoor monitoring by study area.

	Summertime			Wintertime		
Sampling Campaign	Urban Area Median (IQ Range)	Industrial Area Median (IQ Range)	*p*-Value	Urban Area Median (IQ Range)	Industrial Area Median (IQ Range)	*p*-Value
Environmental benzene						
2015 sampling campaign	0.8 (0.7–1.1)	1.1 (1.0–1.2)	0.06	1.6 (1.4–2.1)	3.4 (3.0–4.0)	<0.001
*p*-value vs. summer 2015				0.008	0.008	
2016 sampling campaign	1.0 (0.9–1.2)	1.7 (1.3–2.4)	0.005	n.a.	n.a.	
Environmental toluene						
2015 sampling campaign	3.0 (2.7–3.9)	2.5 (1.9–3.0)	0.072	3.6 (2.8–4.3)	7.8 (6.9–8.1)	0.001
*p*-value vs. summer 2015				0.575	0.008	
2016 sampling campaign	2.7 (2.3–3.0)	2.5 (2.2–3.8)	1.000	n.a.	n.a.	
Environmental T/B						
2015 sampling campaign	3.3 (3.1–3.8)	2.2 (1.9–2.3)	<0.001	2.1 (1.7–2.4)	2.1 (1.8–2.3)	0.970
*p*-value vs. summer 2015				0.005	0.260	
2016 sampling campaign	2.5 (2.4–2.6)	1.6 (1.5–1.7)	0.001	n.a.	n.a.	

**Table 4 ijerph-18-04644-t004:** Airborne benzene and toluene concentrations (expressed as µg/m^3^) in indoor and personal samples by study area.

Samples	Urban Area Median (IQ Range)	Industrial Area Median (IQ Range)	*p*-Value
**Indoor samples**			
**Benzene**	1.5 (0.9–1.6)	1.8 (1.4–2.1)	0.026
**Toluene**	5.0 (4.6–5.4)	5.3 (5.1–7.1)	0.430
**T/B***	4.8 (3.0–6.0)	3.0 (1.7–4.7)	0.024
**Personal samples**			
**Benzene**	1.0 (0.9–1.4)	1.6 (1.4–2.1)	0.001
**Toluene**	3.7 (3.7–7.4)	4.7 (4.7–9.2)	0.070
**T/B***	3.4 (3.4–5.2)	2.8 (2.8–6.5)	0.490

*T/B = Toluene/Benzene Ratio.

**Table 5 ijerph-18-04644-t005:** Median and IQ range of biomarkers of exposure to benzene (U-Benz expressed in µg/L and S-PMA expressed in µg/g creat), cigarette smoke (botinine, ng/gcreat), and oxidative damage to DNA (8-OHdG, µmol/mol creat).

	TOT		Urban		Industrial	
Biomarker	Evening	Morning	Evening	Morning	Evening	Morning
**U-Benz**	0.01	0.01	0.01	0.01	0.02	0.01
Median (IQ range)	(0.01–0.08)	(0.01–0.23)	(0.01–0.02)	(0.01–0.01)	(0.01–0.61)	(0.01–0.59)
**S-PMA**	0.07	0.05	0.07	0.05	0.08	0.06
Median (IQ range)	(0.06–0.11)	(0.04–0.08)	(0.06–0.10)	(0.03–0.06)	(0.06–0.11)	(0.04–0.09)
**Cotinine**	3.12	3.16	3.29	4.25	3.07	2.53
Median (IQ range)	(1.59–5.19)	(1.79–5.48)	(1.72–6.01)	(2.49–6.04)	(1.52–4.95)	(1.39–4.61)
**8-OHdG**	1.12	1.12	1.06	0.98	1.16	1.19
Median (IQ range)	(0.95–1.39)	(0.88–1.34)	(0.90–1.29)	(0.83–1.26)	(1.02–1.51)	(1.03–1.38)

**Table 6 ijerph-18-04644-t006:** Results of the logistic regression analysis with urinary benzene and urinary S-PMA above the upper quartile as the outcome.

Covariates	Urinary Benzene OR (95% CI)	Urinary SPMA OR (95% CI)
**Model 1**		
**Age**	0.9 (0.7–1.2)	0.9 (0.7–1.1)
**Gender (female vs. male)**	0.3 (0.1–1.1)	0.8 (0.3–2.3)
**Area (industrial vs. urban)**	18.5 (1.9–179)	3.2 (0.8–12.5)
**Central address vs. peripheral/rural**	2.6 (0.7–10.5)	2.0 (0.6–6.2)
**Floor level (≥1 vs. street level)**	0.2 (0.04–1.1)	1.9 (0.6–6.2)
***R*^2^**	0.262	0.075
**Model 2**		
**Age**	1.0 (0.8–1.3)	0.9 (0.7–1.1)
**Gender (female vs. male)**	0.5 (0.2–1.7)	0.8 (0.2–2.4)
**Airborne benzene (personal)**	1.4 (0.7–2.6)	3.4 (1.3–8.4)
**Urinary benzene (>fourth quartile)**	-	1.7 (0.4–6.2)
**Smoking cohabitants**	0.5 (0.1–1.6)	1.5 (0.5–4.9)
***R*^2^**	0.037	0.149

**Table 7 ijerph-18-04644-t007:** Results of multiple regression analysis predicting urinary 8-OHdG level.

Covariates	β (95%CI)	Standard Error
**Model 1**		
**Age**	−0.055	0.022
**Gender**	−0.023	0.097
**Area (industrial)**	−0.150	0.100
**S-PMA**	0.331	0.715
***R*^2^**	0.103	
**Model 2**		
**Age**	−0.058	0.022
**Gender**	0.022	0.098
**Area (industrial)**	−0.162	0.106
**Urinary benzene > fourth quartile**	0.012	0.122
***R*^2^**	0.101	

## Data Availability

Restrictions apply to the public availability of these data, the use of which was allowed under license for the current study only. More detailed information access to the rough data is nonetheless possible upon reasonable request and with permission of the Italian Ministry of Health. Please contact Marcello Campagna, Department of Medical Sciences and Public Health, University of Cagliari, for further information.

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
