# Peer review of "Biomarkers of Low-Level Environmental Exposure to Benzene and Oxidative DNA Damage in Primary School Children in Sardinia, Italy"

_ijerph, 2021, doi:10.3390/ijerph18094644_

Round 1

Reviewer 1 Report

This paper studied the impact of low-level benzene exposure on infant’s oxidative DNA damage. This paper was written carefully, but the reviewer don't suggest to be published in Internationa Journal of Environmental Research and Public Health. There are almost half data was below detection limit (see Methodology section and table 5). It indicates that nearly half data were based on assumed value. It is hard to convince the reviewer. Furthermore, other findings in this paper are not new.

Author Response

  1. There are almost half data was below detection limit (see Methodology section and table 5).

In the revised version, we added a paragraph to acknowledge this as a limitation in interpreting our findings. The problem was mainly limited to urinary benzene, and it was due to the low-level environmental exposure. While such a large proportion of benzene readings below the detection limit was an important, reassuring information, we used standard procedures to deal with below LOD values for averting analytical problems in exploring associations with effects at those low levels. Besides, we used a second biomarker of dose, s-PMA, and we explored a binary categorization of benzene and s-PMA excretion.  Both methods yielded substantially consistent results.

  1. …other findings in this paper are not new.

Please see our response to point # 1 raised by the Academic editor:

Epidemiology is an iterative process. Because of the observational nature of epidemiological studies, it is not possible to neutralize all possible confounders and biases through randomization; most such studies have one limitation or another, and final evidence on a given topic results only from their comprehensive evaluation. Therefore, lack of novelty should not diminish the value of a well conducted study, particularly in controversial issues, such as the effect of low-level environmental contamination on children’ health. Also, we respectfully disagree with the Academic editor, as our study differed from most previous publication as it explored the whole pathway from several environmental outdoor and indoor sources of exposure to benzene, to personal monitoring data and biomarkers of exposure, and finally to its effect on DNA damage, using a valuable biomarker of oxidative stress. Most publications only monitored one step, or only measured outdoor or indoor benzene with or without personal sampling and/or biomarkers, and/or only studied oxidative stress. However, we agree with the editor about the need of discussing in more detail the existing literature to highlight the differences with previous work. Therefore, in the revised version, we added several references and extended the paragraph at the end of the discussion by comparing our results to previous publications.

Reviewer 2 Report

The manuscript reports an interesting study about environmental exposure to benzene in children living in an urban area and nearby a petrochemical plant. The results suggested the importance of biological monitoring of low-level environmental exposure and its relation to children's risk of genotoxic effects. I have some comments: How were the children recruited in each school? Why was the number of participants small? Further information on the characteristics of the study city should be described. Environmental assessments and the collection of urine samples were conducted simultaneously in all regions on the same days? Meteorological information should be mentioned. Further details should be added to adequately interpret the results (such as environmental conditions: wind direction, temperature, humidity, a season of the year). Why did the authors choose to perform an analysis of only DNA damage biomarker? For example, the comet and micronucleus assays were not performed.

Author Response

  1. How were the children recruited in each school? Why was the number of participants small?

As explained in the Material and Methods section, the children were recruited in three primary schools of the urban area and in the same number of schools of the town nearby the petrochemical plant. Consistent with similar previous studies, participation was on voluntary base, following meetings with parents in each school to explain the objectives of the study and its protocol. We suspect the complexity of the study protocol, the need for parents to follow precise instructions, and the requirement for children to wear the passive Radiello® sampler all day long, including at school, might have discouraged participation. In discussing the limitations in our study, we included short sentence on this point.

  1. Further information on the characteristics of the study city should be described.

Some characteristics of the two study locations are described in Table 1. In the revised version we added a few more details.

  1. Environmental assessments and the collection of urine samples were conducted simultaneously in all regions on the same days?

As explained in the Materials and Methods section, the three outdoor monitoring campaigns took place in January-March 2015, summertime 2015, and in May-July 2016, the personal and indoor monitoring campaign took place in May-July 2016; parents collected two urine samples of their child the last day of the personal and indoor measurements in May-July 2016.

  1. Meteorological information should be mentioned. Further details should be added to adequately interpret the results (such as environmental conditions: wind direction, temperature, humidity, a season of the year).

As requested, the revised version now includes a paragraph describing average weather conditions in the study areas in the seasons when the two environmental campaigns were conducted. 

  1. Why did the authors choose to perform an analysis of only DNA damage biomarker? For example, the comet and micronucleus assays were not performed.

Please refer to our response to comment # 5 of the Academic editor. Comet and micronucleus assays require fresh leukocytes to be extracted from blood. To promote participation and to match the requests of the Ethical Committee, we refrained from any invasive procedure, such as blood withdrawal, in healthy children, and decided instead to focus on urinary biomarkers of DNA damage at low level environmental exposure.

Reviewer 3 Report

This is an interesting study evaluating the association between different sources of environmental, indoor and personal airborne benzene among 3-13 year children of three locations in Italy. The study investigated if these benzene sources are related to DNA damage caused by oxidative stress. My main concerns are sample size, inclusion criteria and analysis without considering some important confounders such as diet and physical activity. Please see my comments as below:

Page 1, Title: “Biomarkers of low-level environmental exposure to benzene and oxidative DNA damage in children.” This title needs improvement with specifying the age group of children and the place that work has been done. The current title misleads to generalising the outcomes of this study to all children, while this study has been done among 3-13 y children in two locations of southern Sardinia and Sarroch that is a small town and neighbouring one of the largest oil refineries in Europe in Italy.

Page2, line 85, 86: “Children between 3-13 years old, attending primary school in three different locations of either area were eligible for study.” Was this the only inclusion criteria in this study? Why was the study limited to3-13 y children?

-In addition, the authors mentioned  the above age range for children attending primary school. I’m wondering why year 3 of age is considered instead of 6. Were those 3-6 year old children siblings of 6-13 year children from primary schools included in this study? 3-6 y children are not similarly exposed to pollutants. Since the sample size is small and not selected by randomisation this factor can impact the results.

Authors mentioned that dietary habits data was collected in their research in page 3, line 102. While it is strongly evidenced that dietary intakes and nutritional status are associated with urine 8-hydroxy-2'-deoxyguanosine (8-OHdG) level, there is no analysis/adjustment for foods including those with antioxidant effects. Results can be more reliable if nutritional status was considered. What about other lifestyle factors such as intake of fast foods, physical activity and sleeping that can impact DNA damage caused by oxidative stress?

Author Response

  1. main concerns are sample size, inclusion criteria and analysis without considering some important confounders such as diet and physical activity.

As it concerns sample size and inclusion criteria, please refer to our response to comment # 1 of reviewer 2 and to our response # 3 of this reviewer. We are aware of the influence of diet and the microbioma in increasing the production of benzene metabolites, such as phenol, catechol, hydroquinone and t,t-muconic acid, in absence of significant exposure to benzene. However, we relied on benzene and s-PMA as biomarkers of dose, which would be unaffected by diet. As it concerns physical activity, it might increase the resting metabolic rate, but we are unaware of an effect in the metabolism of xenobiotics such as benzene, capable of confounding the relationship between urinary benzene and s-PMA and environmental exposure.  We could not find any publication on physical activity and benzene metabolism, but we are ready to acknowledge the interpretative limitation of having missed considering diet and physical activity as confounders in our study of low-level exposure to benzene in primary school children, if the reviewer would suggest one.

  1. Title: “Biomarkers of low-level environmental exposure to benzene and oxidative DNA damage in children.” This title needs improvement with specifying the age group of children and the place that work has been done.

We changed the title to the following: “Biomarkers of low-level environmental exposure to benzene and oxidative DNA damage in primary school children in Sardinia, Italy”.

  1. “Children between 3-13 years old, attending primary school in three different locations of either area were eligible for study.” Was this the only inclusion criteria in this study? Why was the study limited to 3-13 y children?

We needed collaboration from the participant child in wearing the personal passive sampling device, and therefore children less than 3 years old were not eligible. In the revised version we now clarify that being healthy and not taking any medication was another inclusion criterion.